# Escape from Cellular Senescence Is Associated with Chromosomal Instability in Oral Pre-Malignancy

**DOI:** 10.3390/biology12010103

**Published:** 2023-01-10

**Authors:** Stephen S. Prime, Nicola Cirillo, E. Kenneth Parkinson

**Affiliations:** 1Centre for Immunology and Regenerative Medicine, Institute of Dentistry, Barts and the London School of Medicine and Dentistry, Queen Mary University of London, London E1 4NS, UK; 2Melbourne Dental School, University of Melbourne, 720 Swanson Street, Melbourne, VIC 3053, Australia

**Keywords:** chromosome instability, cellular senescence, tumour development, oral cancer, clinical implications

## Abstract

**Simple Summary:**

Cancer cells commonly escape cell death and proliferate endlessly. We review the molecular mechanisms that are fundamental to this process and suggest that these changes also lead to instability in the genetic make-up (genes and chromosomes) of putative cancer cells. In this way, cells in the very early stages of cancer development acquire unlimited growth potential and also develop cellular diversity, which may facilitate more aggressive cell behaviour and the development of overt cancer. These observations have important clinical implications. We highlight a recent sensitive technique to detect chromosome instability in cells that are shed into the saliva, which can be used for the detection of the early stages of cancer of the mouth. In addition, we discuss new drug developments that are designed to target chromosome instability and, therefore, eliminate potentially dangerous cells before cancer development.

**Abstract:**

An escape from cellular senescence through the development of unlimited growth potential is one of the hallmarks of cancer, which is thought to be an early event in carcinogenesis. In this review, we propose that the molecular effectors of senescence, particularly the inactivation of TP53 and CDKN2A, together with telomere attrition and telomerase activation, all lead to aneuploidy in the keratinocytes from oral potentially malignant disorders (OPMD). Premalignant keratinocytes, therefore, not only become immortal but also develop genotypic and phenotypic cellular diversity. As a result of these changes, certain clonal cell populations likely gain the capacity to invade the underlying connective tissue. We review the clinical implications of these changes and highlight a new PCR-based assay to identify aneuploid cell in fluids such as saliva, a technique that is extremely sensitive and could facilitate the regular monitoring of OPMD without the need for surgical biopsies and may avoid potential biopsy sampling errors. We also draw attention to recent studies designed to eliminate aneuploid tumour cell populations that, potentially, is a new therapeutic approach to prevent malignant transformations in OPMD.

## 1. Introduction

Next-generation sequencing has transformed our understanding of the molecular changes in head and neck squamous cell carcinoma (HNSCC), a term that includes oral squamous cell carcinoma (OSCC). However, cancer genome-sequencing studies have clearly shown that the majority of genetic alterations in a given cancer type are not shared amongst all patients (inter-tumoral) [1], and there is also a striking degree of cell-to-cell genetic heterogeneity within the same tumour (intra-tumoral) [2]. The fundamental driving force of this heterogeneity is thought to be genomic instability [3].

Genome instability describes a spectrum of genetic alterations ranging from small nucleotide changes (mutations, insertions, deletions) to extreme chromosomal alterations. In the present review, we focus on chromosome instability (CIN), which can be defined as an increase in the rate of chromosomal change that manifests as both numerical and structural alterations. By definition, CIN refers to the “rate” of chromosomal instability, and aneuploidy refers to the “state” of chromosomal instability; however, for the purposes of this review, the terms aneuploidy and CIN will be used interchangeably. The literature also refers to somatic copy number alterations (SCNA) and, whilst we are aware that SCNA can be considered as aneuploidy, we propose to continue with the term because it is in common use in the literature. Changes in chromosome number (numerical or whole CIN; W-CIN) are associated with the gains and losses of whole chromosomes and are due to the mis-segregation of chromosomes during mitosis. By contrast, structural CIN (S-CIN) is characterized by an increased rate of change in the chromosome structure such as amplifications, deletions, inversions, duplications and balanced or unbalanced translocations; S-CIN is commonly attributed to errors in the repair of DNA double-strand breaks. Other terms that are commonly used include aneuploidy (the “state” of an abnormal chromosome number, rather than the “rate” seen in CIN), somatic copy number alterations (SCNA) and chromothripsis (genetic chaos).

The impact of CIN in cancer has been described predominantly in terms of advanced tumour progression, drug resistance and clinical outcome [4,5,6]. Early work showed that CIN was also associated with malignant transformation [7,8] but, apart from the fact that CIN has been identified in pre-invasive carcinomas [9], relatively little is known of the causes of CIN and the role of CIN in the early stages of tumour development.

In the present review, we examine the possibility that the development of aneuploidy is closely associated with an escape from cellular senescence. Both CIN and escape from senescence occur early in carcinogenesis; therefore, in this study, we have focused on their role in the malignant transformation of oral potentially malignant disorders (OPMD). Our observations, however, can also be used as a paradigm for epithelial cancers other than OSCC, particularly those of the upper aerodigestive tract and of keratinocyte origin.

## 2. Pathogenesis of OPMD

### 2.1. Clinical Characteristics

HNSCC and OSCC are world health problems and annually account for approximately 800,000 and 350,000 new cases and 450,000 and 170,000 deaths, respectively. OSCC can be preceded by oral potentially malignant disorders (OPMD) that manifest as white (leukoplakia) and/or red (erythroplasia) lesions of the oral mucosa. Neither leukoplakia or erythroplasia can be attributable to any other recognizable condition. The transformation rate for leukoplakia is 2.6–3.5% and for erythroplasia is 14.5–50.0%. The gold standard to assess the potential for malignant transformation is the degree of epithelial dysplasia, and figures of 4.8%, 15.7% and 26.7% are currently accepted for mild, moderate and severe dysplasia, respectively [10]. However, the accuracy of the histological interpretation can be confounded by intra- and inter-observer variation and the presence of reactive epithelial atypia to oral pathogens.

A variety of other oral disorders (oral submucous fibrosis, oral lichen planus, actinic cheilitis, discoid lupus erythematosus, dyskeratosis congenita, Fanconi’s anaemia) also have a propensity for malignant change, but they are associated with disease-specific pathological mechanisms so are not considered further in this review.

### 2.2. Limitations of Current Models of Progression

Historically, the transition to malignancy in epithelial tissues has always been interpreted as being linear. Clinicians, for example, view tumour progression as the transition from normal oral mucosa to OPMD to OSCC, while histopathologists define it as normal oral mucosa to mild/moderate/severe dysplasia to OSCC. From a molecular perspective, neoplastic progression has been described as a continuum of sequential, multiple somatic mutations that lead to the selection of a more dominant cell phenotype. Unfortunately, there is very little evidence to support this view. In the oral environment, for example, not all OPMD are dysplastic; severe dysplasia may remain dormant or even regress; mild dysplasia may progress to cancer; and the clinical appearance of OPMD and OSCC often overlaps making it difficult, if not impossible, to determine whether OPMD has preceded or has occurred concurrently with the development of OSCC. Further, it is now recognized that OSCC can present clinically *de novo*, without being preceded by a precursor lesion. It can always be argued that cancer phenotypes simply progress so quickly through a step-wise morphological continuum that many of the steps are not clinically observable, but the linear theory of cancer development ignores the fact that the phenotypic and genotypic characteristics of putative tumour cells evolve over time [11].

Recently, we reviewed the role of so-called driver genes and tumour suppressor genes in the pathogenesis of OPMD and identified a level of complexity and inter-connectedness not previously described [10]. Despite this information, our capacity to predict malignant transformation in OPMD is limited and has not been translated into clinical practice. This might be explained in terms of a failure to compute sufficient numbers of the predictive factors, threshold levels of key proteins and signalling pathways are never taken into account, and tissue complexity is invariably ignored. However, an alternative view is that the majority of cancer cells are genetically unstable, and it is this instability that leads to the continued evolution of tumour populations. This feature also compounds the difficulty of accurately predicting malignant transformation in potentially malignant disorders.

### 2.3. Somatic Copy Number Alterations (SCNA)

One of the most extensive studies of SCNA in OPMD involved 256 OPMD and 69 paired OSCC [12]. In this study, dysplasia and paired OSCC invariably shared a common ancestry, but approximately one-third of dysplasias had independent SCNA and mutations. Wood and colleagues [12] challenged the commonly held view that the transition from dysplasia to malignancy occurred by way of a random accumulation of genomic changes, because there did not appear to be a step-wise appearance of sub-clones, where each new sub-clone replaced its ancestor. The precancerous field may promote the emergence of a variety of high-risk dysplasias, and, once developed, many lesions could progress simultaneously to OSCC. Wood and colleagues [12] observed that whilst low-grade dysplasia was more likely to have fewer genomic changes than high-grade dysplasia, high-grade dysplasia and OSCC were almost indistinguishable. These authors suggested that the trigger for invasion was more likely to be an additional gene mutation or some other transcriptomic or environmental change.

### 2.4. Driver Genes

Recently, Gerstung et al. [13] used the findings of the Cancer Genome Atlas (2015) to reconstruct the life history and evolution of HNSCC. Primary driver mutations in the development of HNSCC involved TP53, CDKN2A, TERTp, NOTCH1, AJUBA, PIK3CA and CASP8 [13]. This list of driver genes, however, was not particularly comprehensive, because other studies have also identified a broad spectrum of gene abnormalities in OSCC that are likely to contribute to tumour development and progression (FAT1, EGFR, CCND1) [14]. Additional factors [13] are also recognized as being associated with oral epithelial tumour progression, including epigenetic changes, alterations of micro- and long non-coding RNA, involvement of Wnt/β-catenin and NF-κB signalling and changes to DNA damage repair molecules, together with anomalies in immune regulatory cells/molecules and the tumour microenvironment, which are not discussed in the present review.

Abnormalities of TP53 and CDKN2A are common genetic anomalies in OPMD/OSCC. Some 80% of HPV-negative OSCC show inactivation of TP53, either through gene mutation, LOH or increased expression of MDM2; when HPV is present (oropharyngeal cancers), the HPV E6 viral oncoprotein attenuates p53 expression; in these circumstances, TP53 mutations are uncommon. There is a substantial volume of evidence that the inactivation of TP53 correlates with the malignant transformation of OPMD [15,16,17,18,19], but it is cautionary to note that only a small proportion of OPMD progresses to OSCC, despite the prevalence of p53 inactivation (>50%).

p16INK4A inactivation is an important biomarker of malignant transformation in OPMD [20,21,22,23,24,25]. There is evidence that dysfunction of p16INK4A precedes p53 mutations in some OPMD keratinocytes [26], whilst other studies report that it is independent of p53 mutations [27]; such findings suggest that p53 and p16INK4A are regulated differently [28].

## 3. Cellular Senescence

Cellular senescence was originally defined as an irreversible cell cycle arrest that is distinct from quiescence, terminal differentiation and apoptosis, although, more recently, the definition was broadened to include other forms of senescence-related phenotypes [29,30]. It occurs following multiple rounds of cell division (replicative senescence) or in response to a broad spectrum of stresses including DNA damage, oxidative damage, hypoxia, signalling imbalances, activation of oncogenes and cancer-related therapy and ageing [29,30].

### 3.1. Senescence in Keratinocytes

The common effectors of senescence in oral keratinocytes are p53 and pRB/p16INK4A. The p53 tumour-suppressor gene was initially identified as the “guardian of the genome” based on its ability to mediate a G1 arrest following DNA damage. p53 is now known to act in many cellular processes including cell cycle checkpoints, DNA repair, senescence, apoptosis, angiogenesis and surveillance of genome integrity. p53 responds to DNA damage and either activates p21WAF1/Cip1 to initiate cell cycle arrest for DNA repair or, if there is irreparable DNA damage, induces apoptosis [31]. In this way, genomic stability is maintained. Activation of p53 is associated with an increase in its half-life, together with conformational changes mediated by phosphorylation [32,33] and other post-translational modifications [34]. As a result of these functions, p53 has been regarded as a suppressor of gene amplification [35] and aneuploidy [36]. By contrast, CDKN2A is a gene that encodes the cyclin-dependent kinase inhibitors p16INK4A [37] and p14ARF [38]. p16INK4A binds to and inhibits the kinase activity of CDK4/6 and prevents RB phosphorylation [37]. RB remains associated with transcription factor E2F1, thereby preventing the transcription of E2F1 target genes that are essential for transition through the G1-S phase of the cell cycle [39]. By contrast, p14ARF stabilizes p53 and leads to either senescence [38,40] or cell cycle arrest via a p53- and p21CIP1/WAF1-dependent mechanism [41]. However, the induction of G2 cell cycle arrest by p14ARF ultimately leads to cell death through apoptosis [42].

### 3.2. Escape from Cellular Senescence

Senescence has been observed in premalignant lesions of several cancer types [43,44], where it acts as a suppressor of early malignant changes [45]. Later, senescence has the capacity to promote tumour development [46], tumour progression [45,47], tissue plasticity [48] and stem cell activation [49] through the production of the senescent associated secretory phenotype (SASP). Parkinson and colleagues have argued strongly that escape from cellular senescence through inactivation of TP53 and CDKN2A, together with the emergence of telomerase activity, leads to the development of the immortal phenotype in human keratinocytes [50,51,52,53,54,55]. These genetic changes and telomerase deregulation are near ubiquitous event in HNSCC in vitro [51] and in vivo [1].

Disabling p53 extends the proliferative lifespan of fibroblasts but not keratinocytes, whereas knockdown of p16INK4A has no effect on its own. However, the combined knockdown of both p53 and p16INK4A induces a phenomenon that resembles crisis in both cell types [56,57]. Interestingly, in keratinocytes cultured in serum-free conditions, p16INK4A accumulates following proliferative exhaustion [45,51] and is associated with a hyper-motile phenotype seen in carcinoma both in situ and in experimental wounding in vitro [58]. This form of senescence is bypassed when cells are cultured on collagen type I, when either p53 or p16INK4A are disabled, and by inhibition of the TGF-β pathway [58].

The question remains as to whether the molecular mechanisms that are associated with an escape from cellular senescence also drive other aspects of cell behaviour, namely, the development of chromosome instability.

## 4. Development of Aneuploidy

### 4.1. Telomeres

Telomeres are repeats of DNA sequences (TTAGGG) at the ends of human chromosomes that, together with their associated proteins, are collectively referred to as the shelterin complex; these structures ensure that the ends of human chromosomes are not perceived as DNA double-strand breaks by the DNA damage-response machinery [59]. Loss of capping proteins causes telomere shortening and inappropriate joining by non-homologous end joining, which produces dicentric chromosomes.

There is a considerable volume of evidence to show that short telomeres are a characteristic of ageing and age-associated diseases (telomeropathies) such as pulmonary diseases, acquired bone marrow failure syndromes, metabolic disorders and neurodegenerative diseases, amongst others [60]. Telomere attrition occurs in parallel with the loss of replicative lifespan, and, therefore, proliferative arrest eventually occurs by way of the activation of the p53-p19ARF and p16INK4A-Rb signalling pathways. In circumstances where these signalling pathways are intact (i.e., telomeropathies), cells with shortened telomeres eventually progress to senescence and/or apoptosis. In malignancy, however, p53 and p16INK4A are commonly inactivated, while telomerase is activated, with the result being that cells are able to survive chromosomal breakage, thereby causing the development of aneuploidy and the amplification, deletion and translocation of cancer-relevant genes [61]. Certain factors are known to cause both cellular senescence and CIN, but the normal paradigm is that CIN leads to cellular senescence [60].

Short telomeres have been demonstrated in OPMD [62,63], in agreement with other findings from a variety of cancer precursor lesions [64,65,66]. More robust data, however, are required to determine the value of telomere length as a biomarker of tumour development in OPMD. Interestingly, short telomeres in peripheral blood leukocytes also predispose patients with oral premalignant lesions to OSCC, presumably as a result of telomere attrition [67].

### 4.2. Telomerase Expression in Squamous Epithelium

The telomerase enzyme complex consists of dyskerin (DKC1), TERT (telomerase reverse transcription), TERC (telomerase RNA) and other telomerase-related genes.

There is substantial evidence that telomerase activity is highest in the dividing cells of the basal layer of squamous epithelia and that it is down-regulated in the suprabasal layers [68,69] by both transcriptional [70,71] and post-transcriptional [71] mechanisms. Interestingly, telomerase not only prevents telomere attrition [72] but also gradually resolves other forms of DNA damage foci located at the telomeres [73,74]. These observations are entirely consistent with other findings that show normal telomere lengths in normal tissue in close proximity to cancers and pre-malignancies, with the latter pathology having short telomeres despite having detectable telomerase. Furthermore, mortal keratinocytes derived from OSCCs have telomeres of a comparable length to those of normal keratinocytes (J Fleming, K Hunter, EK Parkinson, PR Harrison, unpublished data). Taken together, these findings raise the possibility that putative cancer cells may be derived from telomerase-deficient keratinocytes and that, following crisis, telomerase is deregulated, SCNA develops in parallel, and cells progress to malignancy [27,50,51,75].

The corollary of the above data is that telomerase acts as a tumour suppressor, presumably by counteracting the effects of telomere erosion. It has been shown, for example, that aneuploidy induces replication stress, leading to telomeric DNA damage, p53 activation and p53/Rb-dependent premature senescence in human fibroblasts, an effect that is abrogated by telomerase expression [76]. Further, TERC knockout mice appear to be particularly susceptible to cancer [64,77], again re-emphasising the tumour-suppressor effect of telomerase.

### 4.3. Telomerase Expression in Epithelial Malignancy

Mutations in hTERT are reported in approximately 50% of OSCC [78,79]. Similarly, defects of the genes associated with the telomerase complex are reported in oral pre-malignancies, and these mutations are predisposed to OSCC [80], so they may have predictive value. Dorji et al. [81] examined hTERC in OPMD and demonstrated that progression to malignancy occurred in 9 of 10 lesions with hTERC over-expression and in 1 of 20 cases that retained a normal hTERC copy number. There is also evidence that hTERC as well as hTERT can be rate-limiting for telomerase activity [82].

In cell culture models of human cancer, the dysfunction of the p53 and pRB/p16INK4A pathways leads to the bypass of replicative senescence [83] and extensive telomere erosion, which, ultimately, leads to the formation of complex chromosomal abnormalities [84,85]. Amongst other things and in conjunction with TERTp mutations [86], these chromosomal changes very likely contribute to the deregulation of telomerase [84] and the development of the immortal phenotype, which is a hallmark of cancer [87]. There is abundant evidence that telomere erosion [88] and telomerase deregulation occurs in many human cancers, including HNSCC [89], and that it invariably precedes the development of malignancy [53,54].

### 4.4. Telomere Dysfunction Co-Operates with Inactivation of TP53 and CDKN2A

The combined loss of p53 and p16INK4A is virtually ubiquitous in HPV-negative HNSCC cell lines and tumours [1]. With regard to p53, there is a strong correlation between TP53 mutation and CIN in human solid tumours [90,91,92,93,94]. Li Fraumeni syndrome (LFS), where individuals have TP53 germ-line mutations, has been particularly informative in this context. In LFS individuals, medulloblastomas and acute myeloid leukaemias show chromothripsis [95], and fibroblasts show progressive accumulation of aneuploid cells [96,97], in a similar way to normal human and murine fibroblasts with exogenous expression of TP53 missense mutations [98,99,100]. Further, LFS individuals who develop cancer have an increased germ line copy-number variation [101], suggesting that aneuploidy following TP53 loss of function leads to tumourigenesis. Taken together, studies in LFS indicate that TP53 mutation leads to the development of aneuploidy. The data are entirely consistent with early studies [102] and with the recent analysis of 21,633 TCGA tumour samples across 33 cancer types (including HNSCC), which show a close correlation between TP53 mutation and CIN [103].

Intriguingly, Elmore et al. [104] showed that telomere shortening, rather than the loss of p53 function, was accountable for chromosome instability in Li Fraumeni syndrome, whereas p53 inactivation triggered cellular immortalization. These observations are consistent with the finding that inactivation of p53 and p16INK4A in oral premalignant and malignant cells results in only minor chromosomal alterations but, following telomerase deregulation and cellular immortalization, extensive SCNA and LOH are observed [50,75], suggesting that the emergence of cells from crisis [27] is a key and rate-limiting step in tumour progression. Evidence to support this hypothesis comes from reports that a combination of telomerase deficiency and short telomeres coupled with p53 haplo-insufficiency results in widespread carcinoma development and extensive SCNA in mice [59], some of which are syntenic with those seen in humans [105].

TP53-associated gain-of-function phenotypes (changes in gene expression, clonal growth in vitro, tumourigenicity, metastatic capacity) have been identified in vitro and in vivo, and these cells are invariably aneuploid. Redman-Rivera et al. [106] recently extended these observations and demonstrated that the gain of function phenotypes is independent of p53 alterations and correlates with increased aneuploidy.

With respect to p16INK4A, there is limited information linking CDKN2A inactivation with the development of aneuploidy. However, multiple centrosomes arise in RB compromised cells [92], and the sequential loss of p15INK4B and p16INK4A leads to centrosome duplication, CIN and enhanced proliferation in non-immortalised human cells [107,108].

### 4.5. CIN and Driver Genes Other than TP53 and CDKN2A

NOTCH1 is the second-most-frequently mutated gene in HNSCC after TP53. Loss of NOTCH1 leads to changes in p21WAF1/Cip1 and Wnt/beta-catenin signalling, which result in a decrease in keratinocyte differentiation, increased numbers of keratinocytes in the stem cell compartment [109], alterations in cellular senescence [110,111,112] and changes in epithelial integrity with the development of a wound-like environment [113]. These observations support findings in mice, where targeted deletion of NOTCH1 results in epidermal hyperplasia and the development of skin tumours [114]. Consistent with this tumour-promoting role is the fact that NOTCH1 mutations occur with increased frequency during the ageing of oesophageal epithelium and are exacerbated by ethanol and smoking [115]. However, what may be key to this tumour-promoting activity is that oncogenic NOTCH induces polyploidy mitosis and depolyploidization that result in SCNA and aneuploidy [116]. These findings are consistent with the fact that NOTCH activation is associated with tetraploidy and enhanced chromosomal instability in meningiomas [117]. Possible sources of CIN, SCNA and LOH in squamous epithelium are depicted in Figure 1.

Epidermal growth factor receptor (EGFR) is a receptor tyrosine kinase that binds ligands of the EGF family and activates several signalling cascades to convert extracellular cues into mitogenic responses. EGFR over-expression is common in OSCC [118], correlates with tumour progression [119] and is associated not only with gene amplification but also with multiple centromere copies of chromosome 7, due to centromere region amplification [120]. These findings are consistent with early work demonstrating trisomy of chromosome 7 in non-malignant bronchial epithelium from lung cancer patients and individuals at a high risk for lung cancer [121].

The CCND1 gene encodes cyclin D1, a member of the highly conserved cyclin family. Cyclins function as regulators of specific kinases (CDK4/CDK6), with activity that is required for G1/S transition. The amplification of CCDN1, leading to cyclin D1 over-expression, is common in OSCC (15–55% of tumours), correlates with oral epithelial tumour progression [75] and is associated with both the non-diploid status of tumours [122] and with CIN [123]. It has been suggested that the amplification of EGFR and CCND1 are coordinated and together have an additive effect in the progression of OSCC [124], but this so-called coordination requires experimental verification.

Phosphatidylinositol-4-5-biphosphonate 3-kinases (PI3K) are a family of enzymes that play a pivotal role in transducing the signals involved in a diverse group of cellular functions including cell growth, differentiation, motility, survival and intracellular trafficking. The PI3K/AKT/mTOR pathway is activated by tyrosine kinase receptors such as EGFR and can be antagonized by PTEN (phosphatase and tensin homologue). In HNSCC, the pathway is activated by the oncogenic mutation of PIK3CA, PTEN inactivation through mutation or post-translational modification or over-expression of EGFR [125,126]. The pathway is activated in oral dysplasias showing progression to OSCC [127,128,129]. Emerging evidence indicates that activation of the PI3K pathway can induce and/or allow cells to tolerate CIN [130]. Recently, Zhang and Kschischo [103] showed that high levels of PI3K expression are associated with PIK3CA gene amplification, rarely co-occur with somatic mutations of PIK3CA and PTEN and are associated with high S-CIN. Further, deletion of PTEN leads to aneuploidy and tumour development [131,132], which is perhaps not surprising, since PTEN is involved in the maintenance of genomic integrity [133].

As far as we are aware, there are no published data linking abnormalities of FAT1, AJOUBA and CASP8 with genomic instability.

### 4.6. Spindle Assembly Checkpoints (SAC)

SAC alterations are common causes of aneuploidy and, recently, over-expression of BUBR1 and Mad2 in OPMD and are associated with malignant transformation independent of histological grade [134]; however, the sample numbers are relatively low (n = 52) in this study. Moreover, mutations in SAC genes are yet to be reported in HNSCC, and no functional mutations in BUBR1 are identified in an early study [135].

### 4.7. Anaphase Bridges

Anaphase bridges are DNA threads stretching between two sister chromatids. They arise during DNA replication at times of stress [136,137] and facilitate DNA cohesion. Normally, the bridges are resolved during the S phase, but some persist and can be identified using DNA dyes (Hoechst stain) during anaphase. If they are not removed, however, they lead to gross chromosomal rearrangements and loss of genetic material from at least one of the daughter cells when the bridge eventually breaks [59,138], which, invariably, leads to cell death. In the cells that survive, aneuploidy ensues due to unbalanced chromosomal alterations and chromosomal non-disjunctions [83]. However, the deregulation of telomerase occurs in parallel and results in telomere addition [139] and this, in turn, results in a reduction in the number of dicentric chromosomes and anaphase bridges [84,140,141]. The result is the bulk population of cells escapes crisis and becomes immortal, thereby facilitating tumour progression [64,84,140,141].

Interestingly, post-crisis cancers with de-regulated telomerase appear to be continuously evolving. For example, both established cell lines and tumours in vivo still display a low frequency of anaphase bridges (approximately one in six/seven anaphases [64,141], and, even in HeLa cell populations, clones of cells that have low telomerase activity and shortened telomeres can enter crisis; only when the cells up-regulate telomerase do they start proliferating again [139]. Parkinson and colleagues showed that similar phenomena are present in OSCC lines and tumours [141]. Thus, immortal tumour cell populations seem to exist in a balanced state that favours both continuous proliferation and constantly evolving SCNA.

Recognition of the dynamic state of the tumour genotype has important implications. Studies of the clonal heterogeneity of tumours based on extensive single cell sequencing and SCNA reflect only a “snapshot” of tumour evolution, and, therefore, the data are likely unreliable for the development of new therapeutic modalities. Further, the assessment of malignant transformation in OPMD using genetic criteria might be improved if it was undertaken before the lesions entered crisis, although this, in turn, is challenging because of the heterogeneity inherent in any biopsy sample.

## 5. Function of Aneuploidy

### 5.1. Tumour Promotion

Aneuploidy is considered to be an enabling hallmark of cancer [87] and is a near ubiquitous characteristic of malignancy. It promotes the cancer phenotype (increased cell proliferation, decreased differentiation and apoptosis), overcomes functional defects (promotion of anchorage independent growth, immune escape) and creates genetic complexity, which leads to further evolutionary selection and the development of a “mutator” phenotype that favours proto-oncogene expression and the loss of tumour suppressor genes [142]. Taken together, aneuploidy promotes the cancer phenotype [143,144,145] and leads to enhanced proliferation and spontaneous immortalization [146], altered metabolism [147], transcriptional reprogramming [148], immune evasion [149], migration [150] and invasion and metastases [151].

### 5.2. Tumour Suppression

Despite these traditional views, current thinking indicates that aneuploidy can also suppress malignant cell growth [150,152]. In these circumstances, cellular “fitness” appears to be damaged, leading to a reduction in cell growth and the development of metabolic and proteotoxic stress. Single-chromosome gains, for example, suppress tumourigenesis [146,150,153]. The contradiction between aneuploidy’s capacity to act as a promoter and suppressor of malignant cell growth is described as the aneuploidy paradox [154]. To explain this anomaly, aneuploidy-coping mechanisms are proposed, which include changes in gene expression and the accommodation of proteotoxicity and metabolic stress [155,156]. Indeed, extra copies of the TP53 gene are reported to reduce cancer incidence in mice whilst ageing normally [157].

Recent data show that the situation may be more complicated than first thought. The degree of aneuploidy [158,159], the site of aneuploidy [158] and the genetic background under which aneuploidy operates [160] appear to be important in defining oncogenic potential.

## 6. Does Aneuploidy Initiate Cancer?

Aneuploidy frequently arises early in epithelial cancer [66], but whether it is an initiator of tumorigenesis is unclear [161]. In humans, there is a paucity of information to support this hypothesis, although patients with variegated aneuploidy syndrome are predisposed to childhood cancers [162], and mathematical modelling in colorectal cancer supports a role for chromosomal instability as a tumour initiator [163].

Most of what is known is derived from mouse models of CIN [164], and, because there is an increased incidence of spontaneous tumours in these animals, it has been concluded that aneuploidy is sufficient to initiate cancer. However, the mouse tumour phenotypes in these studies show incomplete penetrance: tumours typically only emerge after long latency periods and, in some models, require a carcinogen to induce tumour formation. Further, the molecular abnormalities that induce aneuploidy in these animal models, namely, defects of spindle assembly checkpoints or their downstream components, are extremely rare in human tumorigenesis. Therefore, what might be more important in tumour initiation are the spectrum of proto-oncogenes that are activated and the array of tumour suppressor genes that are lost as a result of aneuploidy.

## 7. Keratinocyte Cell Lines from OPMD

Parkinson and colleagues [50,51,52,53,54] showed that keratinocyte cultures from OPMD are either immortal or mortal in vitro. The former cell type is characterized by the loss of TP53 and CDKN2A, telomerase activation and extensive SCNA and LOH. In contrast, mortal keratinocytes have wild-type TP53 and CDKN2A pathways and minimal genetic abnormalities. Nevertheless, these OPMD-derived genetically stable keratinocytes can be distinguished from normal keratinocytes by their resistance to suspension-induced terminal differentiation and an altered transcriptional profile [50,51,52,53,54,165]. Taken together, the data show that inactivation of TP53 and CDKN2A, activation of telomerase and extensive SCNA and LOH are associated with an escape from senescence and the development of an immortal phenotype.

Veeramachaneni et al. [75] examined SCNA in the OPMD keratinocyte cultures that were derived by Parkinson and colleagues, showing that those cells that originated from lesions that were known to progress to OSCC (D19, D20, D35) had −3p and −8p with homozygous deletion of FHIT (3p14.1) and CSMD1 (8p23.2). In two of these cultures (D19, D35), there were +3q, +5p, +7p +20 and −13p, −13q, −18p, −18q, +20. In contrast, OPMD keratinocytes derived from lesions that did not progress to OSCC (D34, D4, D9, D38) showed only focal SCNAs involving FHIT (2/4 cultures), CSMD1 (3/4 cultures) and gains of chromosome 20 (3/4 cultures). These results are consistent with the findings of Wood et al. (2017), who examined SCNA in cells from low- and high-grade oral dysplasias; keratinocytes from tissues with a greater propensity for malignant change showed −3p, +3q, −4p, +5p, −5q, +7p, +7q, +8q, −9p, −11p, −11q, +12p, −18q. The biological significance of these chromosome anomalies is yet to be determined.

Using a microcell-mediated chromosome transfer, Vasudevan et al. [160] explored the impact of inducing different trisomies of individual chromosomes on metastatic behaviour. Trisomy of chromosome 5 promoted the metastatic capacity of HCT116 colon cancer cells through the partial induction of EMT. Trisomies of chromosomes 3, 8, 18 and 21 had only minor effects in invasion and cell motility assays, and trisomy of chromosome 13 inhibited invasion in vitro. In OPMD, therefore, the duplication of chromosome 5p and the loss of 13p and 13q may facilitate the invasion of putative tumour cells, and, in these circumstances, the induction of EMT facilitates cells escaping from the boundaries of the basement membrane and the establishment of a primary tumour. Vasudevan et al. [160] concluded that aneuploidy could act as either a tumour promoter or suppressor, but they were cautious in their interpretation of the data because the changes described for HCT116 were not seen in TERT-transformed retinal pigment epithelial cells, suggesting that the effect of specific aneuploidies was context-dependent and related to the genetic and epigenetic background of the target cells. It is also notable that HCT116 shows mismatch binding defects that lead to DNA-damage tolerance [166], which may influence the observed phenotype.

In a parallel approach, de Boer et al. [27] examined the genetic profile of keratinocytes from the surgical margins of OSCC; whilst this is a recognised approach to study cancer development, particularly when investigating field carcinogenesis, the presence of malignant cells in the tissue samples cannot be excluded. The genetic picture was varied (9/27 cultures had TP53 mutations; 8/13 cultures had inactivation of CDKN2A; there was SCNA in the absence of somatic mutations; 2/27 cultures showed complete absence of SCNAs and this was unrelated to the outcome of the tissue of origin) [27]. We combined our own SCNA data [73] with the findings of de Boer et al. [27], and the data show that (1) SCNA was uncommon in mortal keratinocytes (24/32 cultures had no SCNA; 3/32 cultures had only one chromosomal alteration) and that (2) the presence of SCNA appeared to be linked to inactivation of TP53 and CDKN2A. An incidental finding was that NOTCH1 mutations were associated with inactivation of CDKN2A (3/5 cultures). Our interpretation of these observations is that the acquisition of major SCNAs in non-tumourigenic cultured premalignant keratinocytes does not usually occur until cell crisis in vitro is bypassed and telomerase is activated. With regard to events in vivo, escape from cellular senescence through inactivation of TP53 and CDKN2A, plus activation of telomerase, parallels the evasion of crisis in vitro.

## 8. Hypothesis: The Role of Genetic Instability in the Development of OSCC

The concept that cancer evolution follows Darwinian guidelines and leads to the somatic selection of tumour cells has been questioned recently [167]. Vendramin et al. [167] argued that tumour development and progression, for example, are catastrophic events rather than gradual changes, and both neutral evolution and the role of ageing are likely to play a role in carcinogenesis [167]. Further, recent evidence suggests that the spectrum of chromosomal changes that are associated with aneuploidy is not random but shows reproducible patterns during tumorigenesis [159,168].

In Figure 2, we propose a hypothesis to account for the role of genetic instability in the development of OSCC. We suggest that during the premalignant stage of OSCC development, namely, OPMD, putative cancer cells become progressively immortalised by the inactivation of both TP53 and CDKN2A and the activation of telomerase. The result is that these cells not only have an unlimited growth potential but also are able to survive chromosomal breakage, leading to the development of aneuploidy and the amplification, deletion and translocation of cancer-relevant genes. Further, aneuploidy leads to the development of cell diversity and the evolution of cell clones that have the capacity to invade the underlying connective tissue and form primary tumours. If aneuploidy can be considered as a catastrophic event, our proposal is consistent with the work of Vendramin and colleagues [167]. However, we acknowledge that a variety of factors, including the genetic composition of the tumour cells [169,170] and the tumour environment [171,172], amongst others, influence the complex process of invasion.

## 9. Concluding Remarks

The current methodology to assess chromosomal instability has practical difficulties [173] and, therefore, is of limited value in the prediction of malignant transformation. Further, surgical intervention to obtain the appropriate tissue/cell samples is uncomfortable, invariably precludes regular follow-up and, often, is not genetically representative of the tissue of origin. Recently, a new PCR-based assay called the repetitive element aneuploidy sequencing system was described [174], which detects aneuploidy in as little as 3pg of DNA, can be used in a broad range of liquid biopsies and has a sensitivity of 80% when combined with data from somatic mutation analysis and standard protein biomarkers [174]. Similarly, aneuploidy and gene-mutation profiles were used recently to detect malignant nerve sheath tumours in plasma [175]. These techniques lend themselves to the detection of malignant change in OPMD, particularly as saliva analysis abrogates the need for a surgical biopsy.

A characteristic feature of CIN is drug resistance [4], and, therefore, it is particularly challenging to develop drugs that eliminate aneuploid cells [176]; this situation is compounded by intra-tumoral heterogeneity [2]. Nevertheless, appropriate drug development is likely to be rewarding not least because CIN is a near-ubiquitous characteristic of malignant human tumours, though it is extremely rare in normal tissues. At the moment, there are no drugs available in clinics that can be used specifically to inhibit chromosome errors [177], though advances are starting to be made. Aneuploid cells, for example, show increased perturbation of the core components of the SAC; restoration of KIF18A, a mitotic kinesin, leads to a normal response to SAC inhibition [178]. Further, Zhang and Kschischo [103] showed that cells with W-CIN are susceptible to treatment with a BRAF inhibitor (PLX-4032), and those with S-CIN respond to Afatinib, Lapatinib and Austocystin. It remains to be determined whether these drugs will be clinically effective in the control of tumours and potentially malignant disorders in vivo.

Three further approaches also might be used to detect the vulnerabilities of aneuploidy cells: a wide range of drug sensitivity screens, the use of immune checkpoint inhibitors to determine why aneuploid tumours induce weak immune responses and the use of whole genome cDNA, CRISPR and RNAi libraries to modulate gene expression and examine the effect on cancer genotypes [150].

## Figures and Tables

**Figure 1 biology-12-00103-f001:**
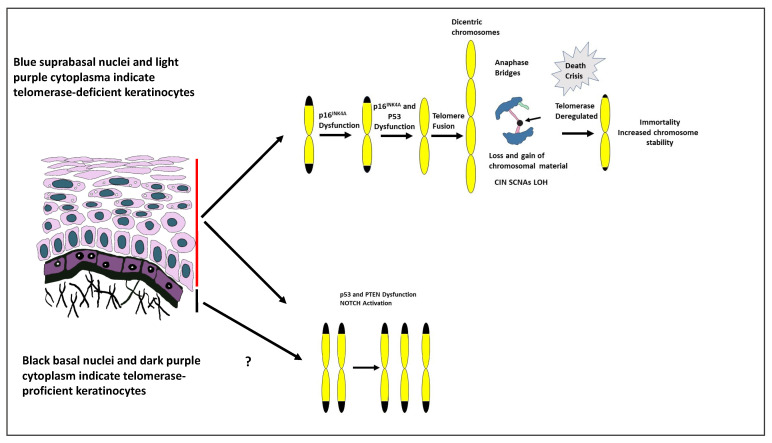
Possible sources of CIN, SCNA and LOH in squamous epithelium. The schematic of squamous epithelium takes into account the many reports that telomerase activity is confined to the proliferative compartment of the tissue and that telomere attrition is minimal in normal tissue. The suprabasal keratinocytes (blue nuclei, light purple cytoplasm) are telomerase-deficient, whereas the basal keratinocytes (black nuclei, dark purple cytoplasm) have sufficient telomerase activity to largely suppress telomere attrition and genomic instability; the latter occurs via chromosome end fusions (fusion–bridge–breakage cycles). It is proposed that neoplastic lesions evolve and progress from telomerase-deficient suprabasal cells and generate SCNA during cellular crisis, prior to immortalization and telomerase de-regulation. Implicit to this process are the end-to-end fusion of chromosomes and the formation of anaphase bridges. In anaphase bridges that link anaphase plates (dark blue), the arrow indicates the fused chromosome ends (black). CIN, chromosome instability; SCNA, somatic copy number alterations; LOH, loss of heterozygosity.

**Figure 2 biology-12-00103-f002:**
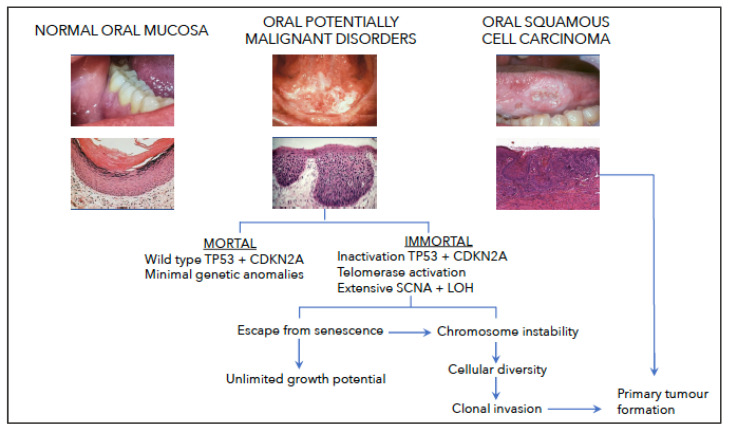
Hypothesis to account for the importance and consequences of genetic instability during the premalignant stage in the development of oral squamous cell carcinoma. The proposal is that during the clinical (normal oral mucosa to OPMD to OSCC) and histological (normal oral mucosa to dysplasia to invasive OSCC) development of oral squamous cell carcinoma, putative cancer cells become progressively immortalised by the inactivation of both TP53 and CDKN2A and the activation of telomerase. The result is that these cells not only have an unlimited growth potential but, also, are able to survive chromosomal breakage leading to the development of aneuploidy and the amplification, deletion and translocation of cancer-relevant genes. Further, aneuploidy leads to the development of cell diversity and the evolution of cell clones that have the capacity to invade the underlying connective tissue and form primary tumours.

## Data Availability

Not applicable.

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
