# Peer review of "Escape from Cellular Senescence Is Associated with Chromosomal Instability in Oral Pre-Malignancy"

_biology, 2023, doi:10.3390/biology12010103_

Round 1

Reviewer 1 Report

Dr. Prime and colleagues review an association between the escape from cellular senescence and chromosomal instability (CIN) in oral pre-malignancy. The review is focused on cellular and chromosomal aspects of specific malignancy, which is truly valuable for specialists in the field. Major omissions are lacking in the review. However, there are several mandatory recommendations, which are to use for making manuscript less limited or less strict in terms of the meaning of cell senescence in CIN formation and consequences.

Firstly, the review creates wrong readers’ opinion that escaping from cell senescence is the only way to CIN associated with cell senescence. Actually, there is a reverse mechanism: cell senescence produces exhausting of mitotic machinery and/or abnormal regulation of cell cycle, which results in CIN, basically manifesting as aneuploidy.  For references, see:

https://doi.org/10.1016/j.mad.2011.10.009;

https://doi.org/10.1016/j.mad.2016.03.007;

https://doi.org/10.3389/fgene.2019.00892;

https://doi.org/10.3390/cells10051256;

https://doi.org/10.1038/s41418-021-00764-5.

Avoiding mentioning this mechanism is unacceptable, inasmuch as it makes understanding of cell senescence contribution to CIN incomplete. For instance, authors describe paradoxical (mutually exclusive) role of aneuploidy in cancer. Similarly, cell senescence has also such a kind of paradoxical (mutually exclusive) role in CIN occurrence.

Secondly, authors have ignored aging as a cause of cell senescence. For, instance aging is required to be included in the list written in lines 160-162. Furthermore, other parts of review truly missing the info about aging and its relation to cancer and CIN.

Thirdly, the method mentioned is not CIN-specific; it is aneuploidy-specific. Accordingly, this fact should be stated, inasmuch as it is impossible to detect chromosomal breakage, fragile sites and other morphological alterations to chromosomes by this technique. Moreover, the technique is poorly described (e.g. control methods, effectivity etc.). One may recommend to exclude mentioning of the technique or to describe it in more details.

Fourthly, Figure 2 requires separate chapter/review part for the description. It is certainly poorly related to concluding remarks. I suggest that the description of this figure is better to designate as a hypothesis for a relationship between escape from cellular senescence and CIN during the development of oral squamous cell carcinoma. It will make the review sounder or more important for basic cancer research and future CIN descriptions.

Finally, some terms require narrowing or substitution. W-CIN is improper term/abbreviation. In fact, authors discuss the effect of aneuploidy only. Polyploidy is not discussed. Accordingly, W-CIN is to substitute by aneuploidy. Formally, aneuploidy is also SCNA. Thus, authors should specify what kind of SCNA they mean. Genetic chaos is not limited to chromothripsis, but also includes chromoanasynthesis, chromoanagenesis and chromohelkosis. Moreover, this term covers almost all types of instable genome behavior and its consequences. It is to avoid the term “genetic chaos” in this given context.

Author Response

Comments attached

Reviewer 2 Report

My comments are included in the enclosed PDF of the manuscript. While the manuscript is well written, it would be useful to include better illustrations i.e figures 1 and 2, with better clarity if possible.

Author Response

Referee 2 has highlighted specific phrases/sentences on the pdf. At these sites, we have addressed any “nit-pik” errors and where appropriate, we have modified the English grammatical style.

Round 2

Reviewer 2 Report

Nicely written review. However,  Figure#1 needs to be revised. The telomeres are within the chromosome and are not an external entity as the figure gives a mistaken impression. I am referring to the black ellipsoid structure shown at the end of each chromosome. Would be useful to include this color within the confines of the yellow boundary of the chromosome itself and show it receding as a consequence of p53/p16 dysfunction.

Author Response

Thanks, we have modified Figure 1